# Research Progress of the Ion Activity Coefficient of Polyelectrolytes: A Review

**DOI:** 10.3390/molecules28052042

**Published:** 2023-02-22

**Authors:** Aokai Zhang, Xiuling Yang, Feng Yang, Chunmei Zhang, Qixiong Zhang, Gaigai Duan, Shaohua Jiang

**Affiliations:** 1Changzhou Vocational Institute of Industry Technology, Changzhou 213164, China; 2Co-Innovation Center of Efficient Processing and Utilization of Forest Resources, College of Materials Science and Engineering, Nanjing Forestry University, Nanjing 210037, China; 3Institute of Materials Science and Devices, School of Materials Science and Engineering, Suzhou University of Science and Technology, Suzhou 215009, China; 4Department of Pharmacy, Sichuan Academy of Medical Science & Sichuan Provincial People’s Hospital, Personalized Drug Therapy Key Laboratory of Sichuan Province, School of Medicine, University of Electronic Science and Technology of China, Chengdu 610000, China

**Keywords:** polyelectrolyte, thermodynamics, nonideality, activity coefficient, chemical potential

## Abstract

Polyelectrolyte has wide applications in biomedicine, agriculture and soft robotics. However, it is among one of the least understood physical systems because of the complex interplay of electrostatics and polymer nature. In this review, a comprehensive description is presented on experimental and theoretical studies of the activity coefficient, one of the most important thermodynamic properties of polyelectrolyte. Experimental methods to measure the activity coefficient were introduced, including direct potentiometric measurement and indirect methods such as isopiestic measurement and solubility measurement. Next, progress on the various theoretical approaches was presented, ranging from analytical, empirical and simulation methods. Finally, challenges for future development are proposed on this field.

## 1. Introduction

Polyelectrolytes are long-chain molecules with ionizable groups. They can dissociate into charged polymer chains and counterions in polar solvents, such as water. Due to combined macromolecular natures and long-range electrostatic interactions, they show impressive properties and possess a wide applications in biomedicine [1,2,3], water clarification [4,5], soft robotics [6,7,8] and energy harvesting [9,10,11,12,13]. Furthermore, natural biological polymers such as DNA and proteins are also polyelectrolytes [14]. The ion activity coefficient stands for the nonideality of the ions in solutions, indicating the chemical reactivity, and allows for the calculation of many thermodynamic properties. It is a fundamental property for biology and geoscience, and also important in technological applications. However, our understanding of the polyelectrolyte activity coefficient is still poor because of the difficulties present in both experiments and theory, and the elegant scaling theories that are successfully developed for neutral polymers are not easily extended to charged systems [15].

Research methods on thermodynamic properties of polyelectrolyte are mostly derived from those of its low-molecular-counterpart, the simple electrolytes. However, significant deviation exists from thermodynamic properties from simple electrolytes. For example, the activity coefficient is known to approach unity for highly diluted electrolyte solutions, whereas the value for polyelectrolyte systems is much smaller. This deviation may ascribe to interactions between the polyelectrolyte chain and counterions, where counterions bind to polyelectrolytes more readily than monomeric electrolytes. A number of comprehensive reviews have covered the areas of thermodynamic properties of simple electrolytes [16,17,18], as well as the solution properties of polyelectrolytes [19,20,21]. Nevertheless, to the best of our knowledge, a review focusing on the activity coefficient of polyelectrolyte is still missing.

Therefore, in this review, the progresses on the activity coefficient of polyelectrolyte systems are summarized (Figure 1). Experimental measurement methods can be roughly divided into two categories, the direct method and indirect ones. The former mainly refers to the potentiometric method, while the latter includes the isopiestic method, solubility determination, and so on. To elucidate the mechanism of extraordinary properties of polyelectrolytes, scientists have made efforts to develop analytical theories to predict the activity coefficients in this kind of system. However, most of these models only take into consideration the long-range electrostatic interactions, and hence can only be used in highly diluted solutions. To solve this problem, empirical terms are introduced to account for other effects, such as short-range interactions. Furthermore, computer simulation enables the controlled design and direct observation of this kind of system, thus may act as an important complement of the other theoretical methods.

## 2. Experimental Determination of Ion Activity of Polyelectrolytes

The activity coefficient is defined as the ratio of the chemical activity of its molar concentration, and the chemical potential of substance *B* is given by:*μ*_B_ = *μ*_B_^⊖^ + *RT*ln*a*_B_ = *μ*_B_^⊖^ + *RT*ln*x*_B_*γ*_B_(1)
where *μ*_B_^⊖^ is the chemical potential of pure *B*, *a*_B_ is the activity of *B*, *x*_B_ is the mole fraction and *γ*_B_ is the activity coefficient.

Activity coefficients of ions in polyelectrolyte systems can be measured from various methods. Potentiometric (electromotive force, EMF) measurement is a direct method to measure the chemical potential, and thus the activity coefficient can be calculated thereafter. It is usually conducted on a potentiometer (or ion/pH meter) in conjunction with an ion-sensitive glass electrode calibrated with solutions of a known concentration of salt (Figure 1a). The Nernst equation is the base to calculate the activity of coefficient *γ*:(2)$$E=E^{0}+\frac{RT}{F}\ln\left[m_{s}\left(m_{s}+m_{p}\right)\gamma_{\pm }^{2}\right]$$
where *F* is the Faraday constant, *E*^0^ the standard potential of the cell, and *m*_p_ and *m*_s_ are the concentrations of polyelectrolyte and salt, respectively. *E*^0^ is determined by EMF measurements in solutions of known salt activity (or calculated by Pitzer’s empirical model) or by extrapolation to infinite dilution.

Kagawa, Ise and co-workers have conducted a series of pioneering EMF studies on polyelectrolyte solutions and found that the mean activity coefficient of polyelectrolytes decreased with increasing concentration [23,24,25,26,27]. The activity coefficients of NaCl in aqueous poly(diallyl dimethyl ammonium chloride) (PDADMA-Cl), sodium salt of poly(anethole sulfonic acid) (NaPASA) and sodium polyacrylate (NaPA) solutions were determined by EMT over the concentration range of 0.01~1 mol kg^−1^. The mean activity coefficient of NaCl was found to decrease monotonically as the salt concentration increased [28]. Sodium ion activity of poly(maleic acid) (PMA) and an alternating copolymer of isobutylene with maleic acid (PIM) in salt-free solution were measured by EMF with a varied degree of dissociation (0.4~1.0) and polymer concentration (2.4~60 mN). The activity coefficient results were compared to the theory prediction [29]. Kienzle–Sterzer and co-workers have conducted EMF measurement on the cationic chitosan. The counterion activity coefficient was found to decrease with a decrease in ionic strength, whereas it remained relatively constant with polymer concentrations [30].

Polyelectrolyte hydrogels are charged polymer networks with ions fixed on polymer chains [31]. The properties of polyelectrolyte gels at the swelling equilibrium are quite different from polyelectrolyte solutions at the same concentration, mainly because counterions confined in gels need to exert an osmotic pressure to balance the elastic energy of polymer network, thus increasing the complexity in such systems [32]. Recently, the EMF method has been extended to polyelectrolyte networks with some modification in apparatus because traditional ones are only suitable for solutions. Safronov and co-workers have studied the activity of counterions in poly(acrylic acid) (PAA) and poly(methacrylic acid) (PMAA) hydrogels by measuring the Donnan potential with the use of capillary electrodes. Then, the activity of free counterion Ag inside the hydrogel could be calculated from the Donnan potential and the activity in the external solution with the equation:(3)$$\phi =\frac{RT}{zF}ln\frac{a_{s}}{a_{g}}$$

The activity of potassium counterions firstly increased and then decreased with the addition of salt for both PAA and PMAA hydrogels [33]. Guo and co-workers have reported, for the first time, the quantitative measurement of local electric potential of brittle polyelectrolyte gels using the microelectrode technique (MET) (Figure 1b). By using microelectrodes with a tip wall thickness less than the local Debye length, the Donnan potential of poly(2-acrylamido-2-methylpropanesulfonic acid) (PAMPS) hydrogel was accurately measured [22]. Recently, with the help of the same technique, Kurokawa and co-workers have successfully estimated the average activity coefficient of the hydrogel and found a difference in concentration between the dense and sparse phases [34]. In addition to the EMT method, it was shown that the activity coefficient of individual ions in polyelectrolyte systems could also be extracted from potentiometric titration data [35,36].

Compared to the potentiometric measurements, other methods are mostly indirect methods. It is possible to calculate the electrolyte activity coefficient *γ*_±_ from the osmotic coefficient *φ*, by applying the Gibbs–Duhem relationship [16]:(4)$$\ln\left(\gamma_{\pm }\right)=\varphi -1+\int_{0}^{m}\frac{\varphi -1}{m}dm$$

Therefore, the activity coefficient can be indirectly obtained from the activity coefficient of the solvent or the osmotic coefficient. For this purpose, several methods may be applied, such as measurements of solvent vapor pressure, freezing point depression, osmotic pressure, among others. With a history of more than 100 years, the isopiestic method still remains one of the most satisfactory techniques for accurate determination of vapor pressures of solutions [37,38,39]. Figure 2a is a diagram of a typical isopiestic apparatus. When performing the experiment, the desiccator is evacuated for equilibrium by transferring solvent mass through the vapor phase. Once the equilibrium is reached, all phases share the same activity. Thus, the osmotic coefficient of the sample can be calculated from the known solutions. The mean activity coefficients of solutes can be derived from isopiestic data based on the Gibbs–Duhem relationship. Ise and co-workers applied the isopiestic vapor pressure measurements and found that the calculated mean activity coefficients were consistent with those previously measured by EMT (Figure 2b) [40]. With this method, the authors also investigated systems with various counterions (salt) types. The difference in activity coefficients was ascribed to diverse structural influences of the ions on water [41,42,43,44]. Compared to the equipment of EMT measurement, the usual isopiestic apparatus is not commercially available. In addition, the relatively long time needed for equilibrium may be a shortcoming of this method. 

For organic liquids, the aqueous activity coefficient is equal to the inverse of the mole fraction solubility [45]. Considering that the most frequently used polyelectrolytes have almost unlimited solubility in water, Dolar and co-workers have chosen barium polystyrenesulfonate (BaPSS) as a test polyelectrolyte with added salt at various concentrations. The calculated mean activity coefficients decreased strongly with increasing salt centration, and were validated by comparison with theoretical values [46]. Apart from the methods mentioned above, freezing-point depression [47], vapor pressure osmometry [48,49] and membrane osmometry [50,51,52] can also measure the osmotic coefficient, and thus derive the activity coefficient in the polyelectrolyte system. It should be noted that the measurability of the individual activity coefficient still remains an open question: some determined it experimentally by, for example, ion-selective electrode [53]. Nevertheless, others denied its measurability for the reason that an individual ion cannot exist independently in a solution [54,55]. The experimental methods are summarized in Table 1.

## 3. Theoretical Models for Ion Activity of Polyelectrolytes

### 3.1. Analytical Model

Most of the theoretical approach to derive the activity coefficient of polyelectrolyte is based on the Poisson–Boltzmann (PB) theory [59], which is a classical mean-field theory that describes the electrostatic interactions in electrolyte solution. The PB theory concerns only Coulombic interactions between point-like charges, neglecting ion correlation and steric effects [60,61,62], thus will apply only to a low concentration. Despite the long history of ionic activity theory since the seminal paper of Debye and Hückel in 1923 [63], a simple physical model for concentrated solutions remain elusive. The improvement of PB theory to adapt to more environments has been studied for years [64,65]. Predominantly, two kinds of models have been used to describe the polyelectrolyte, namely the spherical model [66,67] and the cylindrical model [68] (Figure 3). In the former case, polyelectrolytes are approximated as charged spheres penetrable to small molecules and ions; while in the latter case, polyelectrolytes are modeled as rod-like cylinders. In both cases, the PB equation is solved to calculate the electrostatic potential. In the spherical model, long chain polyelectrolytes were expressed by a rigid sphere with the same charge. The PB equation was solved with certain approximation and the electrical free energy was obtained by a charging-up procedure similar to that of Debye and Hückel for the simple electrolyte system. By differentiating the free energy, they obtained the activity coefficient of the counterion. The theory showed excellent applicability when compared to the experimental data of Ag·CMC solutions [69,70].

The spherical model has the deficiency that it cannot take into consideration the long chain effect of polyelectrolyte, hence the applicability is limited. Katchalsky and Lifson established the fundamentals of the thermodynamic modeling of aqueous polyelectrolyte solutions. In their model, polyelectrolytes take a cylindrical shape because the presence of electric charges stretches the polymer chains. Charges are assumed to be evenly distributed on the cylinder surfaces. The counterions form an ionic cloud around the polymer chain and the electrostatics in the cloud is described by the PB equation [71,72]. The PB cylindrical cell model for salt-free polyelectrolyte solutions accounts well for the thermodynamic properties at low polyelectrolyte concentrations. However, it usually fails at higher concentrations because it neglects the steric effect and ion correlation.

Manning proposed the counterion condensation theory, which is originally introduced by Oosawa’s two-phase model [73]. It is one of the most important extensions of the Katchalsky–Lifson model. In Manning’s theory, a dimensionless parameter is used to measure the polyelectrolyte charge density:(5)$$\xi =e^{2}/\epsilon kTb$$
where *e* is the elementary charge, *ε* is the dielectric constant of the solvent, *k* is the Boltzmann constant, *T* is the absolute temperature in kelvins and *b* is the average spacing distance between charge groups on the polymer.

There exists a critical charge density, *ξ*_crit_, above which counterions condense on the polyelectrolyte backbone to lower the net charge density. This phenomenon is called “counterion condensation”. The PB equation is solved considering the condensation, and the Debye–Hückel approximation is assumed to apply for interactions of uncondensed ions. Apart from polyelectrolyte solutions, Manning’s condensation theory has also been applied to calculate the activity coefficient in the ion exchange membrane. Without adjustable parameters, Manning’s model predicted ion activity coefficients in charged membranes finely (Figure 4a) [74,75]. Similar to PB theory, counterion condensation theory is a limiting law valid only in highly dilute solutions. 

Gueron has derived simple expressions for ion activity coefficients in polyelectrolyte solution from PB theory, based on the assumption that the activities of simple ions are equal to their concentrations at the cell boundary where the electric field is nil. Without including the inclusion of counterion condensation, their results fit the experimental activity coefficient of counterions fairly well. However, neither the condensation theory nor the formulas derived from PB fits the co-ion measurement. Therefore, the authors suggested that the experimental data cannot be called upon to support condensation theory [77].

Among the liquid state integral equation theories, the hypernetted chain equation (HNC) [78] and mean spherical approximation (MSA) [79] also seem useful to describe the ionic distribution. MSA theory is also based on Debye–Hückel theory, while the short-range interactions are expressed by including particle sizes. Maribo–Mogensen has thoroughly compared the differences of MSA theory and DH theory in terms of the numerical results of activity coefficients [80]. Within the binding MSA theory, Bernard proposed a simple model for charged ring polyelectrolyte to describe their thermodynamic properties. The mean activity coefficients and the osmotic coefficient could be calculated by differentiation of the Helmholtz energy [81]. Solms and Chiew presented MSA theory for polyelectrolyte solutions, where polymers were modeled as freely tangent-jointed hard-sphere chains and counterions as hard spheres embedded in a continuum dielectric medium. The mean activity coefficient was derived as a function of system density and Bjerrum length [76]. The combined HNC and MSA equation is utilized to calculate the thermodynamic properties of a polyelectrolyte solution based on the cylindrical cell model. Here, the osmotic and activity coefficient were derived and showed satisfactory agreement with Monte Carlo simulation data. This work indicated that HNC/MSA approximation may be an improvement over the PB equation, especially for di-valent counterions [82]. Another study utilizing HNC/MSA approximation modeled polyelectrolyte solutions as freely tangent-jointed, charged hard-sphere chains. The colligative properties, such as osmotic coefficients and activity coefficients, were calculated for models with various chain lengths. The results agree well with molecular dynamics simulation and experiment results [83].

### 3.2. Empirical Model

Most models for aqueous solutions of polyelectrolytes summarized above are only valid at dilute solutions because they only account for long-range interactions. Pitzer’s model [84] provides a simple and powerful approach to the prediction of activity coefficients and other thermodynamic properties for simple electrolytes [85,86]. It introduced long-range interactions from classical Debye–Hückel theory, while short-range interactions were included via semi-empirical parameters. The activity coefficient is thus given by the sum of terms due to long-range and short-range interactions, respectively:(6)$$ln\gamma =ln\gamma^{LR}+ln\gamma^{SR}$$

Pessoa Filho and Maurer have extended Pizter’s model for polyelectrolyte systems by assuming that the degree of dissociation of polyelectrolyte does not depend on its concentration and modifying the expression for the ionic strength. Ion interaction parameters were correlated with the minimum of adjustable parameters. The predicted osmotic coefficients agree excellently with the experimental data in a wide concentration range [87,88]. In a similar study by Ghalami-Choobar, interaction parameters were evaluated using the Pitzer graphical method. With these parameters, they calculated the mean activity coefficients of NaPA. The agreements between calculations and experiments were satisfactory [89]. Furthermore, Lammertz and co-workers extended the virial equation with the relative surface fractions (VERS) model of Großmann et al. [90,91,92] for neutral polymers and simple electrolytes to account for aqueous solutions of polyelectrolytes and low molecular weight electrolytes, and derived the activity coefficient [93]. Similar to the models of Pessoa Filho and Maurer, the VERS model is also based on Pitzer’s equation. In addition, the ion activity coefficient of natural and synthetic polyelectrolyte has been successfully modeled by using a combination of Pitzer’s model, specific ion interaction theory, and the ion pairing model [94].

An empirical extension has been made to the Manning theory to take account of the interactions between mobile ions. The mean activity coefficient of the salt at the same concentration of co-ion in the absence of polyelectrolyte was added to the contribution of polyelectrolyte–ion interactions calculated from Manning theory, based on the assumption of additivity of excess free energy. The modified Manning theory showed excellent agreement of γ_±_ of NaCl in the presence of sodium dextran sulfate [95]. Nagvekar and Danner introduced a model which is another important example for an extension to higher solute concentrations. In this model, Manning’s concentration law was used to account for long-range interaction, while a local composition model of the non-random two-liquid (NRTL) type was introduced to describe short-range interactions. The excess Gibbs free energy was expressed as the sum of the contribution from long-range and short-range interactions. The model fit the experimental activity coefficients well [96]. Chen and co-workers have established a NRTL model where the long-range polyion–ion interactions were treated with Manning’s limiting law, the long-range ion–ion interactions treated with the Pitzer–Debye–Hückel equation, and the electrolyte NRTL model [97] took care of the short-range molecule–molecule, molecule–ion and ion–ion interactions. The model was capable of systematically correlating the extrapolating activity coefficients of polyelectrolyte solutions [98]. This model was recently extended for multivalent polyelectrolyte systems [99]. It overcame the deficiencies of Manning’s limiting law and correlated activity coefficients of mobile ions successfully.

### 3.3. Monte Carlo Simulation

For polyelectrolytes, analytic theories can only account for the long-range electrostatic interaction in a very approximative way. Mean field theories, as well as statistical thermodynamics, contain simplifications of physical interactions and mathematical approximations, which make the direct comparison with experimental data ambiguous. On the other hand, the polydispersity of synthesized polyelectrolyte imposes a crucial problem in comparison to any theory, which usually assumes a monodisperse case. Computer simulation, such as Monte Carlo simulation and molecular dynamics simulation, is a powerful tool to perform “experiments” under fully controlled condition, and acts as the bridge between analytic theories and experiment [100]. For simple electrolyte, the readers can refer to several comprehensive reviews on the application of computer simulation on the calculation of its thermodynamic properties [17,101]. However, similar reports on polyelectrolytes are missing, to the best of our knowledge.

Monte Carlo (MC) simulation is a sampling algorithm whose idea reaches back to the work of Metropolis and others [102]. MC simulations of polyelectrolytes are usually performed based on the restricted primitive model [103], where charged particles are treated as charged hard spheres, whereas the solvent is modeled implicitly as dielectric continuum [104,105,106]. Widom’s test particle insertion [107] is a simple way to calculate the excess chemical potential of a system of interacting particles, and hence the activity coefficient. The activity coefficient is calculated from the energy difference after or before inserting a “test particle” (Figure 5). It has a great flexibility because it can be implemented in a MC or MD program using various ensembles. Using a spherical cell model, MC simulations were performed for linear polyelectrolytes. The activity coefficients were calculated by Widom’s method with a correction for non-electroneutral insertions into a finite volume [108]. The simulated value agrees well with an expression derived from the Debye–Hückel approximation for a rigid rod with discrete charges at not too high charge density, while Manning’s theory was found to give poor agreement with the simulations [109,110]. Utilizing MC simulation and Widom’s method, Sajevic and co-workers have revealed the effect of chain flexibility on the counterion activity coefficient [111]. When determining the individual ion activity coefficient by Widom’s insertion, one needs to tackle the problem that inserting a single ion may violate the charge neutrality. Recently, Bakhshandeh and Levin discussed two different approaches to solving this problem [112]. Another problem the original Widom’s method faces is the low insertion rate, especially at high density systems [113]. Various methods have been introduced to improve the efficiency of particle insertion [114,115,116,117]. It is noted that in addition to Widom’s method, there are other algorithms, such as Bennets’s acceptance ratio (also known as the overlapping distribution method) [118,119,120] and thermodynamic integration [121,122,123], that are capable of determining the free energy difference, and hence the activity coefficient.

Nishio performed canonical MC simulations of polyelectrolyte solutions with cylindrical cell models. The results of rod-like polyelectrolytes models were compared with the analytical solutions of the PB equation for the uniformly charged cylindrical polyelectrolytes. The counterion distribution agreed with each other [124]. The mean and individual activity coefficients were obtained from the local ion concentration at the cell boundary according to the contact theorem [125]. Compared to experiment data and solutions of the PB equation, good agreements were observed for the mean and counterion activity coefficients. The authors indicated that some reconsideration may be required for the co-ion activity either from the experimental accuracy or from the method of correction [126]. With the same model, the authors also studied the effect of ion size and valence on ion distribution and the osmotic coefficient of rod-like polyelectrolyte solution [127,128].

Compared to traditional canonical MC simulations, MC simulations performed in grand canonical ensemble (GCMC) provide direct access to chemical potentials, and thus to activity coefficients [129,130]. A variation of the primitive model reproduced the experiment results [131]. GCMC simulation results agreed with PB cell model prediction for DNA-salt solutions [132]. A similar study on the DNA-salt system indicated that the PB equation only retains its semiquantitative utility if 1:1 electrolyte is present. Nevertheless, the PB equation does not predict the charge inversion feature observed in a simulated divalent ion system, which results mainly from the neglect of small ion correlation in the PB equation [133]. It is noted that although these works only calculated the mean activity coefficient, GCMC simulation is also capable of calculating individual ionic activity coefficients [134].

### 3.4. Molecular Dynamics Simulation

Molecular dynamic simulation is a brute force approach for computing equilibrium and transport properties of many body systems. It has many aspects similar to real experiments and works by numerically integrating Newton’s equations of motion. Molecular dynamics simulation has been used to study the activity coefficient of simple electrolyte with moderate success [135]. However, similar studies on polyelectrolyte are rare [136]. Rather than the atomistic models, a coarse-grained model is usually used (Figure 6) [137]. In these cases, chemical details of real monomers are ignored. A common example is the model of Kremer and Grest [138,139]. Here, the polymer chain is modeled as mass points which repel each other and are, at the same time, connected along the chain by a spring. The repulsion force, which reproduces the excluded volume constraint, is modeled as the purely repulsive Lennard–Jones (Weeks–Chandler–Andersen (WCA)) potential for good solvent [140]. The spring force takes care of the connectivity. Here, commonly, a finite extensible non-linear elastic potential (FENE) is used. For long-range interactions, the full Coulomb interactions are treated by the Ewald summation method to include interactions of the periodic images. The Coulomb strength is expressed by the Bjerrum length, λ_B_. Particle–particle–particle mesh algorithm, P^3^M [141], is among the most used algorithm that can tackle the calculation of Coulomb interactions.

Antypov and Holm reported an MD simulation on a cell model where the polyelectrolyte was modeled as a finite chain located in the center of a cylindrical cell. The calculated osmotic coefficient correctly reproduced the value of a many-body bulk system up to a dense semidilute regime. The model may be appropriate for stiff-chain polyelectrolytes such as DNA fragments and rigid synthetic polyelectrolytes [142,143,144]. Deserno and Holm developed a Wigner–Seitz cell model for linear polyelectrolytes. The osmotic coefficient, determined from the boundary density, is calculated as a function of concentration and compared to PB theory prediction. They found that for low densities both simulation and PB values converged. Upon increasing the density, the rise in the osmotic coefficient was weaker than the PB prediction, which might be ascribed to enhanced counterion condensation [145,146]. MD simulation of a multichain system with explicit counterions in salt-free solutions has been utilized to calculate the osmotic coefficient of rod-like and flexible polyelectrolytes. These authors found that the osmotic coefficient exhibits nonmonotonic dependence on polymer concentration. In dilute solutions, it decreased with polymer concentration, while it upturned around the overlap concentration [147]. Another multichain simulation study conducted by Stevens and others indicated that the osmotic coefficient has a litter discrepancy between the MD results and those of the HNC/MSA approach [83] in the low-density range [148].

Hybrid MD/MC simulation has also been applied to calculate thermodynamic properties of polyelectrolytes. To access the chemical potential of salt solution in which a polyelectrolyte gel is immersed, a Monte Carlo scheme is incorporated in the Langevin molecular dynamics simulation. In the MC scheme, salt ion pairs are inserted and deleted, and the excess chemical potential is measured. The authors found that, at σ = 0.5 λ_B_, the simulated excess chemical potential agreed reasonably well with both the Debye–Hückel prediction and the experimental data on NaCl [149]. When performing simulations, cares should be taken when setting the simulation box because the calculated activity coefficient is shown to depend on the system size [150]. In addition, the osmotic coefficient/activity coefficient are strongly dependent on the force field [151]. Apart from MC and MD methods, an ab initio quantum chemistry method has also been utilized to calculate small molecules’ activity coefficients with polyelectrolyte membranes [152]. COSMO-RS (conductor-like screening model for realistic solvents), a quantum chemistry-based thermodynamic model, was shown to be a powerful tool for full prediction of activity coefficient properties in fluid mixtures outside the parametrization set [153]. Thus, it essentially brings about the comparison of experimental and theoretical values, and may help to gain more insight about the models of polyelectrolyte systems.

## 4. Perspectives and Conclusions

Despite the increasing experimental and theoretical efforts, the puzzle of the ion activity coefficient in polyelectrolyte systems is far from being completely resolved, even for salt-free cases. The difficulty mainly stems from the complex interaction interplay between the polyelectrolyte, the ions and the solvent. Therefore, combined analytical, empirical and simulation analysis may give more insight into the extraordinary ion activity of polyelectrolytes [154]. In addition, there is still a need for developing new simulation models [155,156] to increase the efficiency and accuracy of activity coefficient calculation. Differing from the introduced direct simulation method, simulation of osmotic pressure provides solvent activity results, from which the salt activity coefficients can be derived using the Gibbs–Duhem equation. To the best of our knowledge, the feasibility of indirect simulation methods has only been proved for low molecular weight electrolyte [157,158]. Thus, its application in polyelectrolyte systems may help to validate the direct simulation and other theories. Compared to relatively mature techniques for solutions, ion activity measurement in polyelectrolyte gels has considerable challenges and may need more improvement in the future. Furthermore, when the concentration increases, there is a need to consider that an ion’s activity may be affected by the change of dielectric constant [159,160,161], which is usually thought to be constant. In addition, discrepancy between experiments and theories is thought to mainly result from imperfections in polymer synthesis, such as the polydispersity of molecular weight and defects of network structure [162,163]. Thus, theories and simulations taking into account these imperfections may provide better prediction of the ion activity coefficient in such systems. Moreover, machine learning may provide a simple pathway to predict thermophysical properties based on the increasing amount of experimental and simulation data [164].

In summary, first, experimental progress was highlighted, with an emphasis on the direct EMF method. Then, theoretical considerations were presented, including analytical methods, empirical equations and simulation approaches. Finally, the review was concluded with a perspective on this area. The physical phenomena of thermodynamics of polyelectrolyte are complicated. Successful prediction of the ion activity coefficient in polyelectrolyte systems calls for careful design of experiments and the application of theoretical models, as well as interdisciplinary collaborations.

## Data Availability

No new data were created or analyzed in this study. Data sharing is not applicable to this article.

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
