# Peer review of "Research Progress of the Ion Activity Coefficient of Polyelectrolytes: A Review"

_molecules, 2023, doi:10.3390/molecules28052042_

Round 1
Reviewer 1 Report
This review article discusses about the methods involved in measuring ion activity coefficient of polyelectrolyte systems. Overall understanding of polyelectrolyte activity measurement will definitely help to gauge mechanisms underlying fundamental property of biological processes. The difficulties in comparing the experimental and theoritical values are efficiently described. Molecular dynamic simulation and Monte-carlo simulation based approaches are explored to some extent. Both direct and indirect method of ion activity calculations are explained but the readers can benefit with additional approaches.
1. For indirect ones, the authors are requested to add on Quantum chemistry calculations. They essentially bring about the comparison of experimental and theoritical values and help to explain the model of the polyelectrolyte systems.
2. For the direct approach, authors can also refer to voltametric titrations or other electrochemical methods.
Author Response
Response to Reviewer 1 Comments
Thank you very much for your kind suggestions and comments. According to the valuable comments of reviewers, we have made the following responses and revisions to our manuscript “Research progress in ion activity coefficient of polyelectrolytes: a review” (molecules-2232047). Corresponding revisions have been marked with red color in the submitted revision.
Point 1: For indirect ones, the authors are requested to add on Quantum chemistry calculations. They essentially bring about the comparison of experimental and theoretical values and help to explain the model of the polyelectrolyte systems.
Response 1: The authors thank the reviewer’s suggestions. Yes, you are right. It is necessary to add the content on the quantum chemistry methods. Hence, in page 11, we have added the following contents
“Apart from MC and MD methods, ab initio quantum chemistry method has also been utilized to calculate small molecules’ activity coefficients with polyelectrolyte membranes [150]. COSMO-RS (Conductor-like Screening Model for Realistic Solvents), a quantum chemistry based thermodynamic models, was shown to be a powerful tool for full prediction of activity coefficient property in fluid mixtures outside the parametrization set. Thus, it essentially brings about the comparison of experimental and theoretical values and may help to gain more insight about the models of polyelectrolyte systems.”
Point 2: For the direct approach, authors can also refer to voltametric titrations or other electrochemical methods.
Response 2: The authors thank the reviewer’s suggestions. In page 2, we have added the following contents on the potentiometric titration.
“In addition to EMT method, it was shown that activity coefficient of individual ions in polyelectrolyte systems could also be extracted from potentiometric titration data [33, 34].”
Reviewer 2 Report
The authors present in this work a review about the research progress in ion activity coefficient of polyelectrolytes. I am very surprised about the high number of references in the article, corroborating that the authors have consulted a wide range of bibliography and verifying the quality of the work. The article is interesting and easy-reading, hence I must accept the article to publish in Molecules, only with minor revision:
1. In line 33, there is a mistake because the words “in biomedical” appear twice. Please, correct it.
2. In line 73, is it “various method” or “various methods”. Please check it.
3. In line 122, what do you refer with osmotic pressure et al.? Is it a mistake?
4. In Table 1, some numbers should appear as sub-index. Please, check them.
Author Response
Response to Reviewer 2 Comments
Thank you very much for your kind suggestions and comments. According to the valuable comments of reviewers, we have made the following responses and revisions to our manuscript “Research progress in ion activity coefficient of polyelectrolytes: a review” (molecules-2232047). Corresponding revisions have been marked with red color in the submitted revision.
Point 1: In line 33, there is a mistake because the words “in biomedical” appear twice. Please, correct it.
Response 1: The authors thank the reviewer’s suggestions. We apologize for the carelessness. The repeated words “in biomedical” is deleted.
Point 2: In line 73, is it “various method” or “various methods”. Please check it.
Response 2: The authors thank the reviewer’s suggestions. We apologize for the carelessness. “various method” is replaced by “various methods”.
Point 3: In line 122, what do you refer with osmotic pressure et al.? Is it a mistake?
Response 3: The authors thank the reviewer’s suggestions. It was a mistake. A comma was missing between “freezing point depression” and “osmotic pressure”. We apologize for the carelessness. Hence, “freezing point depression osmotic pressure, et al.” is replaced by “freezing point depression, osmotic pressure, etc.”.
Point 4: In Table 1, some numbers should appear as sub-index. Please, check them.
Response 4: The authors thank the reviewer’s suggestions. We apologize for the carelessness. The missing sub-index in Table 1 is added.